# Chorioamnionitis and its associated factors among women admitted to the maternity unit of Public Hospitals in Addis Ababa, Ethiopia

Tadios Niguss Derese[1]*, Mekdes Dereje Wondafrash[2], Abel Melese Teka[3], Hiwot Soboksa Mideksa[4], Lidia Dagne Mario[5], Tsegaye Gebreyes Hundie[1], Robel Bayou Tilahun[6], Abel Abebe Demie[7]

1 Department of Research and Training, Eka Kotebe General Hospital, Addis Ababa, Ethiopia, 2 Department of Public Health, Kea Med Medical and Business College, Addis Ababa, Ethiopia, 3 Department of Clinical Pharmacy, Zewditu Memorial Hospital, Addis Ababa, Ethiopia, 4 Department of Internal Medicine, Blacklion Specialized Hospital, Addis Ababa, Ethiopia, 5 School of Public Health, Boston University, Boston, Massachusetts, United States of America, 6 General Practitioner, Uniteam Medical Assistant, Abu Dhabi, United Arab Emirates, 7 Department of Public Health, Gamby Medical and Business College, Addis Ababa, Ethiopia

* Tadiosniguss@gmail.com

## Abstract

Chorioamnionitis is a medical illness marked by maternal fever, leukocytosis, tachycardia, uterine discomfort, and preterm membrane rupture. Chorioamnionitis affects approximately 4% of full-term deliveries worldwide, but it is more common in preterm pregnancies and early membrane rupture. This study aimed to assess the prevalence and associated factors of chorioamnionitis among mothers in public hospitals in Addis Ababa, Ethiopia.A hospital-based cross-sectional study was conducted among maternity patients at Zewditu Memorial, Gandhi Memorial, and Abebech Gobena MCH Hospitals from June 15 to July 15, 2024. A simple random sampling technique with proportional size allocation was used to select a total sample of 379 patients. Data were collected using an interviewer-administered questionnaire and analyzed with SPSS version 26.Binary logistic regression was used to analyze the association between dependent and independent variables. Significance was set at p < 0.05 with a 95% confidence interval, and data were presented in tables and graphs.A total number of 356 patient's data were analysed and the proportion of patients with Chorioamnionitis was found to be 21.3%. After adjustment for possible confounders on multi-variable binary logistic regression analysis age < 25 years [AOR = 0.26, 95% CI (0.09-0.72)], having premature rupture of membrane [AOR = 2.24, 95% CI (1.05-4.78)], duration of labor < 12 hour [AOR = 0.10, 95% CI (0.04-0.24)], and having urinary tract infection [AOR = 3.54, 95% CI (1.72-7.27)] were the significant variable associated with Chorioamnionitis. Chorioamnionitis is a common complication in maternity patients. The magnitude of Chorioamnionitis in the selected public hospital was 21.3%. Age, premature rupture of membrane, duration of labor and urinary tract infection were the factors that had significant association

**Funding:** The authors received no specific funding for this work.

**Competing interests:** The authors have declared that no competing interests exist.

with Chorioamnionitis. The Ministry of Health should focus on creating strategies to prevent and intervene in Chorioamnionitis for patients with premature rupture of membranes and prolonged labor.

## Introduction

Chorioamnionitis, an inflammation of the placenta's membranes and chorion, is primarily caused by bacterial infections following membrane rupture [1]. This condition can occur prior to, during, or following labor and is categorized into acute, sub-acute, or chronic [2]. It poses serious risks to both mothers and newborns, including postpartum infections, preterm birth, and neonatal sepsis [1,2]. Common symptoms include fever, pain in the lower abdomen, and foul-smelling vaginal discharge. Risk factors such as prolonged labor, frequent vaginal exams after membrane rupture, and existing infections increase the likelihood of developing the condition [3,4]. While timely antibiotic treatment can reduce its impact, untreated cases can lead to severe complications, especially in areas with limited access to healthcare resources [5,6].

In Africa and other developing regions, Chorioamnionitis is a significant yet under addressed challenge. Limited access to prenatal care, overcrowded healthcare facilities, and higher rates of infectious diseases exacerbate its impact [7,8]. In Ethiopia, despite government initiatives to provide free maternal healthcare, maternal complications remain alarmingly high, highlighting gaps in prevention and management [9–11]. Addis Ababa, a city marked by its diverse population and stark healthcare disparities, exemplifies these challenges. For many pregnant women here, delayed diagnoses and inadequate treatment options turn a manageable condition into a life-threatening crisis.

This study seeks to address the critical lack of data on Chorioamnionitis in Ethiopia by investigating its prevalence and risk factors among pregnant women in selected maternal clinics in Addis Ababa. By uncovering the extent of the problem and identifying key contributing factors, this research aims to inform targeted interventions that can improve maternal and neonatal outcomes. In a city where healthcare systems are strained and resources are limited, understanding and addressing Chorioamnionitis is not just a medical priority but a moral imperative to protect the lives of mothers and their newborns.

The findings of this study will provide valuable insights to guide prevention, diagnosis, and treatment strategies, ultimately contributing to better maternal health practices and policies in Addis Ababa and beyond. By addressing this understudied issue, the research has the potential to drive meaningful change, ensuring that pregnant women and their families receive the care they need to thrive.

## Methods

### Study area and period

The study was conducted at Maternal Health centers located in Addis Ababa. The study area includes; randomly selected three maternal health facilities in Addis Ababa, Ethiopia (Zewditu Memorial Hospital, Abebech Gobena MCH Hospital and Gandhi Memorial Hospital).

The study was conducted from June 15 to July 15, 2024 Gregorian calendar.

## Study design

An institutional based cross-sectional study was conducted.

## Source and study populations

The source population for this study was all pregnant women attended the maternity unit of public hospitals in Addis Ababa, Ethiopia. The study population for this study was selected pregnant women attended the maternity unit in public hospitals at Addis Ababa, Ethiopia, during the study period.

The sample size for this study was determined using a single population proportion formula considering the assumption

N = the desired sample size

P = Prevalence (p = 34.1%) [12].

Z = is the standard normal score set at 1.96 (95% confidence interval)

D = is the margin of error to be tolerated (5%)

The sample size was calculated using the following single population proportion formula,

n = (z α/2)2 p (1-p)/d$^2$

N = 345, after adding 10% for non-response rate the final sample became 379.

## Sampling technique and procedure

In Addis Ababa, there are eight public hospitals providing maternal health services: St. Paul Hospital Millennium Medical College, Black Lion Specialized Hospital, Zewditu Memorial Hospital, Tirunesh Beijing Hospital, Abebech Gobena MCH, Gandhi Memorial Hospital, St. Peter Specialized Hospital, and Ras Desta Damtew Memorial Hospital. Three hospitals (Abebech Gobena MCH, Zewditu Memorial Hospital, and Gandhi Memorial Hospital) were selected using a simple random sampling technique. Then the number of study units to be sampled from each hospital was determined using proportional to size allocation formula.

$$\frac{ni * nf}{N}$$

For Zewditu Memorial hospital= 450*379/2050=83

For Gandhi Memorial hospital= 750*379/2050=139

For Abebech Gobena hospital= 850*379/2050=157

Where ni = number of maternity patients in each maternity clinic

nf = Final sample of the study

N = total number of maternity patients in the selected public hospitals

Simple random sampling technique was used to select maternity patients from each clinic.

### Inclusion and exclusion criteria

**Inclusion criteria.** All admitted women in the maternity unit of the selected public hospitals during the study period were included in the study.

**Exclusion criteria.** Admitted women in the maternity unit of the selected public hospitals who were critically sick during the time of data collection and those women who were not willing to participate were excluded from the study.

### Study variables

**Dependent Variable:** Chorioamnionitis

**Independent variable**

**Socio-demographic variables:** maternal age, marital status, educational status, occupational status, residence, monthly income.

**Obstetrics variables:** parity, gestational age, preterm delivery, premature rupture of membranes (PROM), abortion, cesarean section (CS) delivery, vaginal examination, antenatal care (ANC), duration of labor.

**Medical and lifestyle related factors:** UTI, alcohol use, tobacco use, chronic disease, fever, and antibiotic use.

## Operational definition

Chorioamnionitis is diagnosed by the presence of maternal fever (temperature >37.8°C) accompanied by two or more of the following criteria: 1) maternal tachycardia (heart rate >100 beats/min); 2) uterine tenderness; 3) foul-smelling AF; 4) fetal tachycardia (heart rate >160 beats/min); and 5) maternal leukocytosis (leukocyte count >15,000 cells/mm$^3$) [13].

## Data collection procedure

A structured questionnaire was adopted from established literature following a thorough review of relevant studies [12–16]. The questionnaire consisted three parts, part I: Patient demographic variables. Part II: medical and obstetrics related variables part III: questions for diagnosing Chorioamnionitis. Three BSC midwives were recruited and participate throughout the data collection and were trained for half day by the principal investigator on the study instrument and data collection procedures. The data was collected by the data collectors through interview.

## Data quality control

Data collectors were selected carefully based on their educational status. After training was given to data collectors to ensure the validity and reliability of the data collection tool, pre-test was done on 5% (nineteen) of the total sample size at Saint Paul Hospital Millennium Medical College (SPHMMC) before the actual data collection and the questionnaires were checked for its clarity, understandability and simplicity. Structured questionnaire was prepared in English and later translated to the Amharic language in which the researcher used it for data collection. The principal investigator checked the collected data, and any incomplete documents were cleaned, checked for quality prior to data entry in SPSS V-26.

## Data analysis

The data was analyzed using SPSS version-26 software. The descriptive analysis was done by simple frequencies and proportions, and the results were presented by tables and words. Binary logistic regression model was used to assess the association between the independent variable and the outcome variable (Chorioamnionitis, yes/no). Bi-variable analysis was performed to screen out potentially significant independent variables at 25% level of significance to be included in the multivariable binary logistic regression analysis. Odds ratio, p-value and 95% CI for odds ratio was used for testing significance and interpretation of results. Variables with p- value of <=0.05 was considered to be statistically significant variable in the multi-variable analysis.

## Ethical consideration

Ethical clearance for this study was obtained from Gamby Medical and Business College Institutional Review Board (Reference number- GMBC/077/2024). Verbal informed consent was obtained from each study participant after the purpose and objective of the study was clearly shared. To document this, we recorded the date, time, and key details of each consent discussion, with an independent witness present to ensure transparency, and this process was approved by the IRB. Participants were also informed that participation is on a voluntary basis and that they could

withdraw from the study at any time if they were not comfortable. For the purpose of confidentiality, the names of participants were not recorded. Except the principal investigator and the research team, no other person would have access to the collected data.

## Result

### Socio-demographic characteristics

A total number of 356 study participants participated in the present study, with a response rate of 93.9%. The mean age of the participants was 28.85 years with a standard deviation of 7.66 years.

Majority of the study participants 273 (76.7%) were married, while the rest 83 (23.3%) were unmarried. Majority 192 (53.9%) of the respondents completed higher education, 90 (25.3%) completed secondary education, 62 (17.4%) completed primary education and the rest 12 (3.4%) were illiterate (Table 1).

### Obstetrics and medical related characteristics of the respondents

Two hundred forty seven (69.4%) of the respondents did not had history of premature rapture of membrane. One hundred seventeen (32.9%) of the respondents had previous history of Cesarean delivery. One hundred fifty six (43.8%) of the respondents had greater than or equal to six vaginal examination, and the rest 200 (56.2%) had less than six vaginal examination. Majority 248 (69.7%) of the respondents did not had urinary tract infection. Majority 164 (46.1%) of the

**Table 1. Socio-demographic characteristics of the respondents (n = 356).**

| Variables | Frequency | Percentage |
|---|---|---|
| **Age** | | |
| < 25 Years | 133 | 37.4 |
| 25-35 Years | 135 | 37.9 |
| > 35 Years | 88 | 24.7 |
| **Marital status** | | |
| Married | 273 | 76.7 |
| Unmarried | 83 | 23.3 |
| **Religion** | | |
| Orthodox | 165 | 46.3 |
| Protestant | 76 | 21.3 |
| Catholic | 45 | 12.6 |
| Muslim | 70 | 19.7 |
| **Educational status** | | |
| Can't read and write | 12 | 3.4 |
| Primary education | 62 | 17.4 |
| Secondary education | 90 | 25.3 |
| Higher education | 192 | 53.9 |
| **Occupational status** | | |
| Government employee | 101 | 28.3 |
| Private employee | 89 | 25.0 |
| Daily labourer | 49 | 13.8 |
| Housewife | 117 | 32.9 |
| **Residence** | | |
| Addis Ababa | 318 | 89.3 |
| Outside Addis Ababa | 38 | 10.7 |

Global Public
Health
PLOS

respondents had less than 12 hours labor duration, followed by 111 (31.2%) 12–18 hours and 81 (22.8%)>= 18 hours, respectively (Table 2).

## Magnitude of chorioamnionitis

The overall Magnitude of clinical Chorioamnionitis in the selected three public hospitals was found to be 21.3% (Table 3).

**Table 2. Obstetrics and medical related characteristics of the respondents (n = 356).**

| Variables | Frequency | Percentage |
|---|---|---|
| **Gestational age** | | |
| <=34 weeks | 105 | 29.5 |
| >34 weeks | 251 | 70.5 |
| **Preterm delivery** | | |
| Yes | 116 | 32.6 |
| No | 240 | 67.4 |
| **Premature rupture of membrane** | | |
| Yes | 109 | 30.6 |
| No | 247 | 69.4 |
| **Abortion** | | |
| Yes | 166 | 46.6 |
| No | 190 | 53.4 |
| **CS delivery** | | |
| Yes | 117 | 32.9 |
| No | 239 | 67.1 |
| **Vaginal examination** | | |
| >= 6 times | 156 | 43.8 |
| < 6 times | 200 | 56.2 |
| **Urinary tract infection** | | |
| Yes | 108 | 30.3 |
| No | 248 | 69.7 |
| **ANC visit** | | |
| Regular | 217 | 61.0 |
| Irregular | 139 | 39.0 |
| **Alcohol use** | | |
| Yes | 151 | 42.4 |
| No | 205 | 57.6 |
| **Current antibiotic use** | | |
| Yes | 146 | 41.0 |
| No | 210 | 51.0 |
| **Duration of labor** | | |
| < 12 hours | 164 | 46.1 |
| 12-18 hours | 111 | 31.2 |
| >18 hours | 81 | 22.8 |
| **Chronic medical disease** | | |
| Yes | 165 | 46.3 |
| No | 191 | 53.7 |

**Table 3. Diagnosis of Chorioamnionitis (n = 356).**

| Variables | Frequency | Percentage |
|---|---|---|
| **Fever>38$^0$c** | | |
| Yes | 87 | 26.4 |
| No | 269 | 75.6 |
| **Maternal tachycardia** | | |
| Yes | 159 | 44.7 |
| No | 197 | 55.3 |
| **Fetal tachycardia** | | |
| Yes | 85 | 23.9 |
| No | 271 | 76.1 |
| **Foul smelling discharge** | | |
| Yes | 59 | 16.6 |
| No | 297 | 83.4 |
| **Uterine tenderness** | | |
| Yes | 42 | 11.8 |
| No | 314 | 88.2 |
| **Leukocytosis** | | |
| Yes | 55 | 15.4 |
| No | 301 | 84.6 |

Among the 131 study participants from Gandhi Memorial Hospital, 28 (21.4%) were diagnosed with Chorioamnionitis. Similarly, of the 148 participants from Abebech Gobena MCH, 33 (22.3%) were diagnosed with the condition. Additionally, out of the 77 participants from Zewditu Memorial Hospital, 15 (19.5%) were diagnosed with Chorioamnionitis (Table 4).

## Associated factors of chorioamionitis

After adjustment for possible confounders on multivariable analysis age, PROM, duration of labor, and UTI have significant association with the outcome variable in multivariate analysis at 95% CI ($p < 0.05$).

Pregnant women who have had PROM were 2.24 times more likely to have Chorioamnionitis than their counter parts [AOR = 2.24, 95% CI (1.05-4.78)]. Urinary tract infection was also another significant variable which had an association

**Table 4. Overall magnitude of Chorioamnionitis and magnitude in different selected hospitals.**

| Name of Hospitals | Frequency | Percentage |
|---|---|---|
| **Abebech Gobena MCH** | | |
| Yes | 33 | 22.2 |
| No | 115 | 77.7 |
| **Gandhi Memorial Hospital** | | |
| Yes | 28 | 21.4 |
| No | 103 | 78.6 |
| **Zewditu Memorial Hospital** | | |
| Yes | 15 | 19.5 |
| No | 62 | 80.5 |
| **Overall magnitude of Chorioamnionitis** | | |
| Yes | 76 | 21.3 |
| No | 280 | 78.7 |

Global Public Health

PLOS

with Chorioamnionitis, respondents who had urinary tract infection were 3.54 times more likely to have Chorioamnionitis than respondents who did not had urinary tract infection [AOR = 3.54, 95% CI (1.72-7.27)]. Pregnant women with age less than 25 years had 74% reduced odds of having Chorioamnionitis than older pregnant women [AOR = 0.26, 95% CI (0.09-0.72)]. Duration of labor was also another significant variable that showed a significant association with Chorioamnionitis [AOR = 0.10, 95% CI (0.04-0.24)] (Table 5).

## Discussion

The socio-demographic characteristics of the study participants provide valuable insights into the population studied. The majority of participants were married (76.7%) and fell within the 25–35 age group (37.9%), reflecting a relatively young and family-oriented cohort. Over half of the respondents (53.9%) had completed higher education, suggesting a relatively high level of educational attainment among the participants. Additionally, nearly 90% of the participants resided in Addis Ababa, indicating that the findings may be more representative of an urban population.

The magnitude of Chorioamnionitis among pregnant women at  Zewditu Memorial hospital, Gandhi Memorial Hospital and Abebech Gobena MCH hospital  was found to be 21.3%. The magnitude of Chorioamnionitis in this study was higher than a study conducted at Mekaneselam primary hospital (3.7%) [17], Dhaka, Bangladesh (12.7%) [18], Al-Wakra, hospital (2.06%) [19]. The magnitude of Chorioamnionitis in this study was lower than a study conducted at Mbarar regional

**Table 5.  Bi-variable and multivariable analysis for chorioamnionitis (n = 356).**

| Variables | Chorioamnionitis Frequency (%) | | COR (95% CI) | AOR (95% CI) | P-value |
|---|---|---|---|---|---|
| | **Yes** | **No** | | | |
| **Age** | | | | | |
| < 25 Years | 32 (42.1) | 101 (36.1) | 0.50 (0.28-0.90) | 0.26 (0.09-0.72) | 0.010 |
| 25-35 Years | 10 (13.2) | 125 (44.6) | 0.12 (0.06-0.27) | 0.05 (0.01-0.14) | 0.000 |
| >35 Years | 34 (44.7) | 54 (19.3) | 1 | 1 | |
| **PROM** | | | | | |
| Yes | 36 (47.4) | 73 (26.1) | 2.55 (1.51-4.30) | 2.24 (1.05-4.78) | 0.035 |
| No | 40 (52.6) | 207 (73.9) | 1 | 1 | |
| **Vaginal examination** | | | | | |
| >= 6 times | 45 (59.2) | 111 (39.6) | 2.21 (1.31-3.70) | 1.88 (0.96-3.70) | 0.065 |
| < 6 times | 31 (40.8) | 169 (60.4) | 1 | 1 | |
| **UTI** | | | | | |
| Yes | 33 (43.4) | 75 (26.8) | 2.09 (1.24-3.54) | 3.54 (1.72-7.27) | 0.001 |
| No | 43 (56.6) | 205 (73.2) | 1 | 1 | |
| **ANC** | | | | | |
| Regular | 55 (72.4) | 162 (57.9) | 1.90 (1.09-3.32) | 1.55 (0.74-3.24) | 0.240 |
| Irregular | 21 (27.6) | 118 (42.1) | 1 | 1 | |
| **Duration of labor** | | | | | |
| < 12 hours | 18 (23.7) | 146 (52.1) | 0.094 (0.049-0.18) | 0.10 (0.04-0.24) | 0.000 |
| 12-18 Hours | 12 (15.8) | 99 (35.4) | 0.092 (0.04-0.19) | 0.09 (0.03-0.22) | 0.000 |
| >18 hours | 46 (60.5) | 35 912.5) | 1 | 1 | |
| **Parity** | | | | | |
| Nulliparous | 43 (56.6) | 135 (48.2) | 1.40 (0.84-2.33) | 1.71 (0.72-4.02) | 0.218 |
| Multiparous | 33 (43.4) | 145 (51.8) | 1 | 1 | |
| **Current antibiotic use** | | | | | |
| Yes | 26 (34.2) | 120 (42.9) | 0.693 (0.40-1.17) | 0.57 (0.28-1.17) | 0.127 |
| No | 50 (65.8) | 160 (57.1) | 1 | 1 | |

hospital, Uganda (34.1%) [12], Uttara Adhunik medical college and hospital, Dhaka, Bangladesh (25.4%) [16], Popayen Colombia (68.2%) [14]. This variation may be due to the difference in sample size, the socio-economic and cultural differences of the study participants. While the result was somewhat expected given the context of Addis Ababa's healthcare challenges, it underscores the urgent need for targeted interventions to address this high burden and improve maternal and neonatal outcomes.

The finding of this study revealed that pregnant women who have had labor duration of less than 12 hour had a less likely chance to develop Chorioamnionitis than women with labor duration of greater than 18 hours. Likewise, this study was comparable with a study done in Mbarara regional referral hospital, Uganda [12], showed that women with a prolonged labor were more likely to develop Chorioamnionitis, a possible explanation for this association is that prolonged labor may increase the risk of bacterial infiltration into the amniotic fluid. Similarly, because the fetal membranes are exposed to the vaginal flora for a longer period of time during prolonged labor, there is a higher risk of infection. Reduced labor lengths could decrease this exposure and lower the chance of developing Chorioamnionitis.

Premature rupture of membrane was also another significant variable, which had a significant association with Chorioamnionitis, pregnant mothers with PROM had a two folds increase in having Chorioamnionitis. This can be explained as; during PROM the amniotic sac is broken for a long time before birth, this lets bacteria in the vagina move up into the uterus. Similar findings were observed in a study conducted at Dhaka, Bangladesh [12].

Pregnant women with urinary tract infection were more likely to have Chorioamnionitis than pregnant women who did not have urinary tract infection. And pregnant women with age less than 25 years were found to be protective for Chorioamnionitis. A possible explanation for this could be that bacteria from the urinary tract, often due to UTIs, might enter the amniotic sac, increasing the chance of infection in the protective membranes around the fetus. Contrary finding were observed in a study conducted at Mbarar regional hospital, Uganda, which revealed that young pregnant women were more likely to have Chorioamnionitis than older pregnant women [12]. The reason for this difference might be due to the variations in population demographics, healthcare access, and cultural practices of the study populations.

The finding of this study also showed that pregnant women with age less than 25 years had 74% reduced odds of having Chorioamnionitis than older pregnant women. This could be because younger women might be more likely to adopt healthier habits or receive more consistent prenatal care, both of which can contribute to better overall pregnancy outcomes. There may be additional social factors at play, such as reduced rates of prior pregnancies or fewer exposure hazards.

These findings translate directly into actionable clinical strategies: implementing stricter monitoring for women with PROM to enable early antibiotic administration, routine UTI screening during prenatal visits with prompt treatment, and developing protocols to identify and manage prolonged labor more aggressively. The protective effect of younger maternal age suggests we should particularly focus these interventions on older mothers. These targeted approaches could significantly reduce chorioamnionitis rates in our setting.

### Limitation of the study

There was no adequate or similar study conducted in Ethiopia and even there was limited documentation so that the discussion was not adequately described. Not following the standard diagnosis for Chorioamnionitis could also be another limitation for this study. And due to the sensitive nature of some questions (like abortion) the information collected might not reflect the truth, but the data collectors tried their best to have accurate information as possible.

### Conclusion

This study, conducted at Zewditu Memorial Hospital, Gandhi Memorial Hospital, and Abebech Gobena MCH Hospital, revealed that Chorioamnionitis remains a significant maternal health concern. Key factors associated with an increased risk of Chorioamnionitis included premature rupture of membranes and urinary tract infections. Conversely, younger maternal age and the absence of prolonged labor were associated with a reduced likelihood of the condition. These

findings underscore the importance of targeted prenatal care strategies, particularly for high-risk groups, to mitigate the burden of Chorioamnionitis and improve maternal and neonatal outcomes.

## Recommendation

The Ministry of Health should focus on creating strategies to prevent and address Chorioamnionitis, especially for patients with premature rupture of membranes or prolonged labor.

The Addis Ababa City Administration Health Bureau should prioritize raising awareness among the community about the importance of preventing urinary tract infections during pregnancy. At the same time, healthcare providers should focus on identifying the root causes of urinary tract infections in pregnant women and make sure they are treated as early as possible.

## Acknowledgments

Thanks go to the data collectors, supervisor, and all the study participants who generously contributed their time to make this paper possible.

## Author contributions

**Conceptualization:** Tadios Niguss Derese, Mekdes Dereje Wondafrash, Lidia Dagne Mario, Abel Abebe Demie.

**Data curation:** Tadios Niguss Derese, Abel Melese Teka, Hiwot Soboksa Mideksa, Tsegaye Gebreyes Hundie, Robel Bayou Tilahun.

**Formal analysis:** Tadios Niguss Derese, Mekdes Dereje Wondafrash, Abel Abebe Demie.

**Investigation:** Tadios Niguss Derese, Mekdes Dereje Wondafrash, Abel Abebe Demie.

**Methodology:** Tadios Niguss Derese, Hiwot Soboksa Mideksa, Tsegaye Gebreyes Hundie.

**Project administration:** Tadios Niguss Derese.

**Resources:** Hiwot Soboksa Mideksa.

**Software:** Tadios Niguss Derese, Mekdes Dereje Wondafrash, Robel Bayou Tilahun.

**Supervision:** Tadios Niguss Derese, Hiwot Soboksa Mideksa, Tsegaye Gebreyes Hundie, Abel Abebe Demie.

**Validation:** Abel Melese Teka, Lidia Dagne Mario, Robel Bayou Tilahun.

**Visualization:** Abel Melese Teka.

**Writing – original draft:** Tadios Niguss Derese, Mekdes Dereje Wondafrash, Lidia Dagne Mario, Abel Abebe Demie.

**Writing – review & editing:** Tadios Niguss Derese, Abel Melese Teka, Hiwot Soboksa Mideksa, Tsegaye Gebreyes Hundie, Robel Bayou Tilahun.

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
