## [Decision Letter · Decision Letter 0]

PGPH-D-25-00381

Chorioamionitis and its associated factors among admitted women in maternity unit of Public Hospitals in Addis Ababa, Ethiopia.

Dear Derese,

Thank you for submitting your manuscript to PLOS Global Public Health. After careful consideration, we feel that it has merit but does not fully meet PLOS Global Public Health’s publication criteria as it currently stands. Therefore, we invite you to submit a revised version of the manuscript that addresses the points raised during the review process.

We look forward to receiving your revised manuscript.

Kind regards,

Collins Otieno Asweto, PhD

Academic Editor

Journal Requirements:

1. In the ethics statement in the Methods, you have specified that verbal consent was obtained. Please provide additional details regarding how this consent was documented and witnessed, and state whether this was approved by the IRB.

Reviewers' comments:

Reviewer's Responses to Questions

**Comments to the Author**

1. Does this manuscript meet PLOS Global Public Health’s publication criteria? Is the manuscript technically sound, and do the data support the conclusions? The manuscript must describe methodologically and ethically rigorous research with conclusions that are appropriately drawn based on the data presented.

Reviewer #1: No

Reviewer #2: Yes

2. Has the statistical analysis been performed appropriately and rigorously?

Reviewer #1: Yes

Reviewer #2: No

3. Have the authors made all data underlying the findings in their manuscript fully available (please refer to the Data Availability Statement at the start of the manuscript PDF file)?

Reviewer #1: Yes

Reviewer #2: No

4. Is the manuscript presented in an intelligible fashion and written in standard English?

Reviewer #1: No

Reviewer #2: Yes

5. Review Comments to the Author

Reviewer #1: TOPIC

Chorioamionitis and its associated factors among admitted women in maternity unit of Public Hospitals in Addis Ababa, Ethiopia.

Remark(s) 1: First, delete the full stop at the end of the topic. Second, reduce the font size from 16 to 12. Third, the title should read: “Chorioamionitis and its associated factors among women admitted to the maternity unit of Public Hospitals in Addis Ababa, Ethiopia”

ABSTRACT

Background:

Remark(s) 2: Purpose: Based on the purpose of the study, there will be need to revise the title slightly to capture the key variables (prevalence and factors) in the purpose.

Remark(s) 3: Results: “a total number of 356 patients were interviewed and the proportion of patients with Chorioamnionitis was found to be 21.3%.” Two issues with this, first, cap the letter “a” starting the sentence. Second, what research approach did you use? Did you actually conduct interviews as portrayed by the phrase “…356 patients were interviewed….”? Consider rephrasing statement to reflect the specific approach used so as not to leave the global audiences in doubt.

Remark(s) 4: Conclusion: Add a sentence on recommendation to the conclusion.

INTRODUCTION

Lines 50 & 51, para 1: Chorioamnionitis is an acute inflammation of the placenta's membranes and chorion, usually induced by an ascending polybacterial infection after membrane rupture.

Remark(s) 5: First, insert page numbers. Second, provide a source for this claim (definition).

Lines 52 & 53, para 1: This infection can occur prior to, during, or following labor and is characterized as acute, subacute, or chronic (2).

Remark(s) 6: Revise to read: “This infection can occur prior to, during, or following labor and is categorised into acute, subacute, or chronic (2).”

Lines 57-60, para 2: Chorioamnionitis is a common…. Group B streptococcus….

Remark(s) 7: Consider merging this with the “para 1” – though two-sentence paragraphs are used, they are not good enough in academic writing.

Lines 74 & 75: There is a lack of comprehensive studies in Ethiopia assessing the prevalence and factors contributing to Chorioamnionitis.

Remark(s) 8: “There is a lack of….” – most scholars have issues with this kind of phrase. Cautious language such as: “To our knowledge, there appears to be a lack of….” is preferred. Consider revising it.

Remark(s) 9: The section will benefit from some bit of revision to make the problem more obvious. The definition and description of the diseases alone has taken about 50% of the entire section. The authors must provide details on how the problem exist in Africa and other developing countries, with a brief comparison to what pertains in the developed world. Why is the condition an issues for Africa/developing countries in general and Addis Ababa, Ethiopia in particular? Provide a stronger justification for the study.

METHODS

Lines 96 & 97: The sample size for this study was determined using a Single population proportion formula Considering the assumption

Remark(s) 10: Use lower case letters for the words flagged in the sentence above.

Inclusion and exclusion criteria

Inclusion criteria

All clients at the selected public hospitals receiving treatment or admitted during the study period were included in the study.

Exclusion criteria

Those who were critically sick during the time of data collection and those who were not willing to participate were excluded from the study.

Remark(s) 11: There are few issues with the “Inclusion and exclusion criteria”. First, you title is specific on women who are on admission at the Maternity Unit of Public Hospitals in Addis Ababa, Ethiopia. Meaning, these hospitals have other units that provide different services. Therefore, your “Inclusion and exclusion criteria” must capture these characteristics (women on admission, maternity unit, public hospitals in Addis Ababa, Ethiopia). Revise the section to reflect your topic.

Lines 134-138: Operational definition

Remark(s) 12: This is not a good fit for the methods; move to “Introduction”.

Lines 139-145: Data collection procedure

Remark(s) 13: Was the instrument developed purposely for this study? Was the instrument adopted or adapted from literature? These questions must be adequately addressed to establish validity and reliability of the instrument.

Data quality control

Remark(s) 14: You indicated that you conducted “pre-test” done for the instrument. Why not pilot-testing? What was the outcome of the pre-test? What is the significant value for the testing? This is supposed to be a purely quantitative study and there are conditions precedent on this approach.

RESULTS

Lines 192 & 193: Pregnant women’s who have had PROM were 2.24 times more likely to have chorioamnionitis than their counter parts [AOR= 2.24, 95% CI (1.05-4.78)].

Remark(s) 15: Two issues, first, to ensure consistency, use initial cap for the diseases (chorioamnionitis) throughout the work. Second, correct this “women’s”.

Remark(s) 16: While the section was detailed enough and covers the full breadth of the title, the authors must indicate their give their professional opinions on the findings. Were the findings expected or not expected? Were the findings surprising based on their knowledge of the diseases? Merely presenting the results does not help much. Remember, giving a brief opinion on the findings is different from the how the work must be discussed.

DISCUSSION

Lines 204 & 205: The magnitude of chorioamnionitis among pregnant women’s at Zewditu Memorial hospital, Gandhi Memorial Hospital and Abebech Gobena MCH hospital was found to be 21.3%.

Remark(s) 17: “…women’s.” This is becoming repetitive…fix this throughout the work.

Lines 205-209: This finding was higher than a study conducted at Mekaneselam primary hospital (3.7%) (17), Dhaka, Bangladesh (12.7%) (18), Al-Wakra, hospital (2.06%) (19). The finding of this study was lower than a study conducted at Mbarar regional hospital, Uganda (34.1%) (12), Uttara Adhunik medical college and hospital, Dhaka, Bangladesh (25.4%) (16), Popayen Colombia (68.2%) (14),

Remark(s) 18: There are few issues with the above. First, second sentence ended with a coma instead of a full stop. Second, what do the phrases “This finding was higher than a study conducted…” and “The finding of this study was lower than a study conducted” mean? How can the findings of study be “higher” or “lower” than a previous study? Revise the above to give meaning to the argument.

Lines 212-214: The finding of this study revealed that pregnant women’s who have had a labour duration of less than 12 hour had a less likely chance to develop chorioamnionitis than women’s with a labour duration of greater than 18 hours.

Remark(s) 19: Fix the portions flagged in red above.

Lines 214-217: …which showed that women’s with a prolonged labour were more likely to have chorioamnionitis (12), the possible explanation for this might be due to prolonged labour may open up the possibility of bacteria getting into the amniotic fluid.

Remark(s) 20: Fix the issues raised above…the sentences do not read well.

Lines 228-230: The possible explanation for this might be due to bacteria from the urinary tract may enter the amniotic sac as a result of UTIs, raising the possibility of infection in the membranes encircling the fetus.

Remark(s) 21: This does not read well…fix throughout the discussion.

STRENGTH AND LIMITATION OF THE STUDY

Lines 242 & 243: The main strength of this study is that, it is not a single center study. i.e. the study conducted at Zewditu Memorial hospital, Gandhi Memorial Hospital and Abebech Gobena MCH hospital.

Remark(s) 22: This is inaccurate – that the study was merely cross-sectional does not pass muster for a strength. You need to revise this. Moreover, the sentence does not read well, recast.

Line 244: And close supervision during the data collection was made by the principal investigator.

Remark(s) 23: Revise the above sentence, it is not a good academic writing style. Additionally, this cannot be a strength, delete and find a better justification.

CONCLUSION

Lines 251-254: The magnitude of chorioamnionitis in a comprehensive study conducted at Zewditu Memorial Hospital, Gandhi Memorial Hospital, and Abebech Gobena MCH Hospital, was found to be relatively high. This significant magnitude highlights the importance of understanding the various risk factors associated with this condition.

Remark(s) 24: The above do not read well. What is the meaning of “This significant magnitude highlights….”, “…magnitude of chorioamnionitis in a comprehensive study conducted….”? Stick to the key constructs in your topic/purpose/aim to draw your conclusion.

Lines 256-258: To the contrary, certain factors were identified as protective against the condition; notably, not experiencing prolonged labor and being of a younger age were associated with a reduced risk of chorioamnionitis.

Remark(s) 25: This does not read well, recast.

Lines 258-261: These findings highlight the significance of targeted interventions and monitoring techniques for at-risk populations, as well as more study to investigate the underlying mechanisms driving these relationships.

Remark(s) 26: This does not read well, recast.

Remark(s) 27: The section was not well composed and does not pass muster for a good conclusion. The authors must rewrite this section completely.

RECOMMENDATIONS

Line 264: Based on the findings the following recommendation were forwarded

Remark(s) 28: What does “recommendation were forwarded” mean? Correct this. Additionally, there is subject verb disagreement, recast. Provide “:” at the end of this statement.

Lines 265-267: Ministry of Health may consider more work in planning and designing strategies to prevent and make intervention on chorioamnionitis, focused on patients with premature rupture of membranes and patients with prolonged labour.

Remark(s) 29: This does not read well, recast.

Lines 268-270: Addis Ababa city administration health office needs to create awareness for the community about urinary tract infection during pregnancy and healthcare providers needs to investigate the root cause of urinary tract infection in pregnant women and need to treat as early as possible.

Remark(s) 30: If you are addressing an organization, then you need to write the name properly. Is this “Addis Ababa city administration health office” the official name? Correct this. Moreover, this is not correct “create awareness for the community”.

Remark(s) 31: The section must be rewritten – it was not well written.

COMMENTS FOR THE AUTHORS

The authors have touched on a topic I surmise will be help inform policy and practice. I commend the authors for this bold decision. However, there are significant gaps which include grammatical errors and poor articulation of thoughts and ideas. These and many other observations captured above make the paper not interesting read. I suggest to the authors to seek assistance from an English Language expert and experienced researcher.

Reviewer #2: Review report: Chorioamionitis and its associated factors among admitted women in maternity unit of Public Hospitals in Addis Ababa, Ethiopia.

Proposed title: Review report: Chorioamionitis and its associated factors among admitted women in maternity unit in selected Public Hospitals in Addis Ababa, Ethiopia.

Methods:

Page 7: Line 89: The study was conducted from June 15 to July 15, 2024 G.C. Write the GC in full for avoiding confusion as its not a known world out of Ethiopia.

Page 8: Line 93:, all clients admitted to..: This word all clients must be changed to all pregnant women attended the maternity of ….

Sampling technique and procedure:

Explain how three hospitals were only selected. How many hospitals with maternity services in Adis Ababa? Was really randomly or purposefully selected? If it was randomly carried out, please elaborate on this, state all public hospitals and the procedure you used to select

Study Variables

Page 9: 126 Dependent Variable: Chorioamnionitis: explain how this variable was obtained. Was there any confirmation test done? Have you set the cutoff of signs and symptoms as you mentioned diagnosis of Chorioamnionitis on the line: 134 Operational definition: Were all have to be fulfilled as citated: explain how you judged the overall results for Chorioamnionitis according to the definition.

Page 9: line 130: Start by writing in full these words such as: PROM

The data quality control:

Page 10: Line 148, 149 …..pre-test was done on 5% : Just put the reference of the extent or percentage of sample size. Is it 5%?

Were other quality control measures (Validity and Reliability test) used?

Page 10: line: 155. Data analysis

Include the measurement of Chorioamnionitis, what the measures used for qualifying it: Yes or No?? or Cutoff?? Confirmation test?? Or other?

Why haven’t you used Chi-square or Fisher test instead of using the same test in bivariate and multivariate? For confounding adjustments.

Could you add a small paragraph of ethical clearance?

Results:

Page 11-16: 172 Table 1, table 2 and table 3. Separate the frequency and the percentage. Include also the totals down to see if no missing numbers.

Page 16: Table 4-Bi-variable and multivariable analysis for chorioamnionitis (n=356). There is no need of frequencies unless it is the bivariate analysis with chi-square or Fisher test. Check if you can use them separately.

Page 16: Line 187: make a separate table showing the overall prevalence of chorioamnionitis and prevalence in different selected hospitals.

Page 18: 203 Discussion Discuss your socio-demographic results.

Conclusion: Avoid discussion in conclusion, and conclude according to your findings

Page 21: 271 Ethical considerations: take this in methodology

Contribution: Put all authors contributions.

References

Just follow the Vancouver referencing style.

6. PLOS authors have the option to publish the peer review history of their article (what does this mean?). If published, this will include your full peer review and any attached files.

**Do you want your identity to be public for this peer review?** For information about this choice, including consent withdrawal, please see our Privacy Policy.

Reviewer #1: **Yes: **BOTHA, Nkosi Nkosi

Reviewer #2: **Yes: **Nsanzabera Charles

---

## [Decision Letter · Decision Letter 1]

PGPH-D-25-00381R1

Chorioamnionitis and its associated factors among women admitted to the maternity unit of Public Hospitals in Addis Ababa, Ethiopia

Dear Dr. Derese,

Thank you for submitting your manuscript to PLOS Global Public Health. After careful consideration, we feel that it has merit but does not fully meet PLOS Global Public Health’s publication criteria as it currently stands. Therefore, we invite you to submit a revised version of the manuscript that addresses the points raised during the review process.

Thank you for revising your manuscript to address the concerns raised by reviewers 1 and 2. Reviewer 1 has evaluated your revised manuscript and is satisfied that you have addressed the concerns raised in the previous round of review. However, I have noted some additional issues with your manuscript that require further revisions:

1) The literature review is brief and does not motivate the study. Please explain why you chose the specific predictor variables that you did and what your hypotheses were regarding the associations between these variables.

2) Age is presented as three categories rather than as a continuous variable. It is not clear why it was trichotomised or why the specific age groups were chosen. Please include age as a continuous predictor in your analyses rather than treating it as categorical.

3) Your interpretation of odds ratios in the results section is incorrect. You describe ORs as changes in likelihoods rather than changes in odds. For example, an odds ratio of 2.24 does not mean that women with PROM are 2.24 times as likely to be diagnosed with Chorioamnionitis as women without PROM. Rather, it can be interpreted as a 2.24 increase in odds. Looking at your data, the actual likelihood/risk of a Chorioamnionitis diagnosis was around twice as great in women with PROM.

4) You state that one of the diagnostic criteria for chorioamnionitis is the presence of maternal fever (temperature >37.8°C), but in table 3 you use >38°C. Which is correct?

5) There is a disconnect between the stated aims of study (lines 69-73) and what was studied/concluded. It is not clear how your findings can "inform targeted interventions" or  "guide prevention, diagnosis, and treatment strategies".

Could you please revise the manuscript to carefully address these concerns?

We look forward to receiving your revised manuscript.

Kind regards,

Steve Zimmerman, PhD

PLOS Staff Editor

Additional Editor Comments (if provided):

Reviewers' comments:

Reviewer's Responses to Questions

**Comments to the Author**

1. If the authors have adequately addressed your comments raised in a previous round of review and you feel that this manuscript is now acceptable for publication, you may indicate that here to bypass the “Comments to the Author” section, enter your conflict of interest statement in the “Confidential to Editor” section, and submit your "Accept" recommendation.

Reviewer #1: All comments have been addressed

2. Does this manuscript meet PLOS Global Public Health’s publication criteria? Is the manuscript technically sound, and do the data support the conclusions? The manuscript must describe methodologically and ethically rigorous research with conclusions that are appropriately drawn based on the data presented.

Reviewer #1: Yes

3. Has the statistical analysis been performed appropriately and rigorously?

Reviewer #1: Yes

4. Have the authors made all data underlying the findings in their manuscript fully available (please refer to the Data Availability Statement at the start of the manuscript PDF file)?

Reviewer #1: Yes

5. Is the manuscript presented in an intelligible fashion and written in standard English?

Reviewer #1: Yes

6. Review Comments to the Author

Reviewer #1: The authors have done well in addressing my recommendations.

7. PLOS authors have the option to publish the peer review history of their article (what does this mean?). If published, this will include your full peer review and any attached files.

**Do you want your identity to be public for this peer review?** For information about this choice, including consent withdrawal, please see our Privacy Policy.

Reviewer #1: **Yes: **BOTHA, Nkosi Nkosi

---

## [Editor Report · Decision Letter 2]

PGPH-D-25-00381R2

Chorioamnionitis and its associated factors among women admitted to the maternity unit of Public Hospitals in Addis Ababa, Ethiopia

Dear Dr. Derese,

Thank you for submitting your manuscript to PLOS Global Public Health. After careful consideration, we feel that it has merit but does not fully meet PLOS Global Public Health’s publication criteria as it currently stands. Therefore, we invite you to submit a revised version of the manuscript that addresses the points raised during the review process.

We look forward to receiving your revised manuscript.

Kind regards,

Shiyam Sunder, MBBS, MSc epidemiology, PhD

Academic Editor
---

## [Editor Report · Decision Letter 3]

Chorioamnionitis and its associated factors among women admitted to the maternity unit of Public Hospitals in Addis Ababa, Ethiopia

PGPH-D-25-00381R3

Dear Derese,

We are pleased to inform you that your manuscript 'Chorioamnionitis and its associated factors among women admitted to the maternity unit of Public Hospitals in Addis Ababa, Ethiopia' has been provisionally accepted for publication in PLOS Global Public Health.

Best regards,

Shiyam Sunder, MBBS, MSc epidemiology, PhD

Academic Editor